# Developmental factors associated with decline in grip strength from midlife to old age: a British birth cohort study

Diana Kuh, Rebecca Hardy, Joanna M Blodgett, Rachel Cooper

MRC Unit for Lifelong Health and Ageing at UCL, London, UK

**Correspondence to**
Professor Diana Kuh;
d.kuh@ucl.ac.uk

## ABSTRACT

**Objectives** To test whether developmental factors are associated with grip strength trajectories between 53 and 69 years, and operate independently or on the same pathway/s as adult factors.

**Design** British birth cohort study.

**Setting** England, Scotland and Wales.

**Participants** 3058 men and women.

**Main outcome measures** Grip strength (kg) at ages 53, 60–64 and 69 were analysed using multilevel models to estimate associations with developmental factors (birth weight, growth parameters, motor and cognitive development) and father's social class, and investigate adult factors that could explain observed associations, testing for age and sex interactions.

**Results** In men, heavier birth weight, beginning to walk 'on time', later puberty and greater weight 0–26 years and in women, heavier birth weight and earlier age at first standing were independently associated with stronger grip but not with its decline. The slower decline in grip strength (by 0.07 kg/year, 95% CI 0.02 to 0.11 per 1 SD, p=0.003) in men of higher cognitive ability was attenuated by adjusting for adult verbal memory.

**Conclusions** Patterns of growth and motor development have persisting associations with grip strength between midlife and old age. The strengthening associations with cognition suggest that, at older ages, grip strength increasingly reflects neural ageing processes. Interventions across life that promote muscle development or maintain muscle strength should increase the chance of an independent old age.

## INTRODUCTION

Maintaining musculoskeletal function for the maximal period of time, and preventing musculoskeletal disorders are important aspects of healthy ageing, enabling people to remain active and independent for longer.[1][2] Worldwide, musculoskeletal disorders rank second in terms of impact on years lived with disability.[2] The role of muscle mass, strength and metabolic function is recognised for these disorders, and is becoming more widely appreciated in cardiovascular and other chronic diseases.[3][4]

Hand grip strength is a commonly used indicator of muscle strength[5] and an overall

### Strengths and limitations of this study

► This British birth cohort study has prospectively assessed factors from development onwards, has a wide range of potential covariates and remains broadly representative of the population born in Britain in the early postwar period.

► There are repeat measures of grip strength assessed over a relatively long follow-up from midlife (53 years) to early old age (69 years), where there have been few studies of age-related decline.

► It was only possible to model linear change as there are currently only three assessments of grip strength, but we did investigate whether each association strengthened or weakened with age.

► While accounting for deaths and attrition, and those unable to provide a grip measurement for health reasons, did not alter our findings, the observed associations could still be explained by residual confounding.

biomarker of ageing.[6–8] Average levels rise to a peak during the early 30s, plateau and then decline.[9–11] Weaker grip is associated with future morbidity, disability and mortality across populations of different ages, ethnicities and income levels,[4][12–18] as is decline in grip strength.[19][20] Adult risk factors, including height and adiposity, health conditions, cognition and health behaviours, have been associated with subsequent grip strength,[21][22] and with age-related decline.[23–34] Developmental factors, such as birth weight, physical growth, motor and cognitive development and childhood socioeconomic conditions are also related to adult grip strength[35–37] but evidence on whether they are associated with age-related decline is limited.[34][38][39] Using two repeat measures of midlife grip strength in the Medical Research Council (MRC) National Survey of Health and Development (NSHD, the oldest of the British birth cohort studies), Cooper *et al* showed that those of higher childhood cognitive ability and socioeconomic position (SEP) (but not higher birth weight) had a reduced risk of decline

in grip strength between 53 and 60–64 years.[34] Findings from the Lothian Birth cohort 1921 showed that during late life (79–87 years) those from a lower childhood SEP had a greater decline in grip strength,[39] but there was no association with childhood cognitive ability.[38]

A research gap is to understand whether developmental factors operate on the rate of decline in grip strength independently or on the same pathway/s as adult factors to inform the timing and types of interventions that may modify this decline. Based on previous findings,[34 36 39] and using three repeat measures of grip strength ascertained from age 53 to 69 years in the NSHD, we tested two hypotheses: (1) higher birth weight, greater gains in height and weight during childhood and adolescence and earlier puberty are associated with a greater grip strength but not its decline; and (2) higher childhood cognitive ability, achieving motor milestones early or around the modal age and higher childhood SEP are associated with greater grip strength and a slower rate of decline. Furthermore, we expected that any associations between indicators of physical growth and grip strength to be explained by adult health conditions and body mass index (BMI), and that any associations between motor and cognitive development, childhood SEP and grip strength to be explained by education and adult cognition.

## METHODS
### Sample
The MRC NSHD is a sample of all births in 1 week in March 1946 in mainland Britain comprising 5362 (2547 female) individuals followed up 24 times, so far to age 69 years.[40] The maximum sample for these analyses comprised 3058 participants with at least one measure of grip strength at ages 53, 60–64 or 69 years and known adult height and birth weight. Of the remaining 2304 in the original birth cohort, 738 had died, 542 were living abroad, 270 had been lost to follow-up and 166 had not provided all necessary data. Those still participating in the NSHD in adulthood have lower childhood SEP, and higher adult SEP and childhood cognitive ability than those who did not, but the sample remains generally representative of those born in Britain in the mid-20th century.[41 42] Participants provided written informed consent for each visit.

### Grip strength
During nurse assessments at ages 53 and 60–64, grip strength was measured in kilograms isometrically using a Nottingham electronic handgrip dynamometer; during a nurse home visit at age 69, a Jamar Plus+ Digital Hand dynamometer was used. A randomised repeated-measurements cross-over trial found no statistically significant differences in values when comparing these devices.[43] We applied the same standardised protocols and used the maximum of the first four measures (two in each hand) at each age.

### Childhood factors
*Birth weight:* birth weight, extracted from birth records to the nearest quarter pound, was converted to kilograms.

*Physical growth:* the Superimposition by Translation and Rotation (SITAR) model of growth curve analysis was used to estimate individual patterns of growth in height and weight between 0 and 26 years.[44 45] Heights and weights were measured using standardised protocols at ages 2, 4, 6, 7, 11 and 15, and self-reported at ages 20 and 26. The NSHD data were augmented by height and weight data between 5 and 19 years from the Avon Longitudinal Study of Parents and Children (ALSPAC) cohort. Subject-specific random effects were obtained for size, tempo and velocity (in SD units) for height and weight.[44] Later puberty is indicated by positive tempo values and earlier puberty by negative values.[44]

*Motor and cognitive development:* age (in months) at first sitting, standing and walking was based on maternal recall at age 2. At age 15, a standardised measure of childhood cognitive ability was derived from the Heim AH4 test of fluid intelligence, the Watts Vernon reading test and a test of mathematical ability.[46 47] Standardised scores (mean=0; SD=1) from similar tests at ages 11 or 8 were used if missing at age 15 as participants maintained similar ranking across time; Pearson's correlation coefficients between the overall cognitive test scores at these three ages were between 0.7 and 0.9.[34]

*Childhood SEP:* father's occupation at age 4 (or at age 11 or 15 if missing at age 4), based on the Registrar General's Social Classification, distinguished three groups: high (I or II), middle (IIINM or IIM) and low (IV and V).

### Adult factors
*Height and adiposity:* at ages 53, 60–64 and 69, height (cm) and weight (kg) were measured using standard protocols and BMI (kg/m$^2$) was calculated; standardised scores were used in analyses.

*Health conditions:* at age 53, a summary of health conditions was a count (0–4) of the presence of knee osteoarthritis, hand osteoarthritis (both based on clinical assessment), severe respiratory symptoms and other potentially disabling or life-threatening conditions.[25]

*Education and verbal memory:* highest educational attainment by age 26 distinguished those with a degree or higher, advanced secondary, ordinary secondary, other or no qualifications. At ages 53, 60–64 and 69, verbal memory was assessed using a 15-item word list task over three trials (range 0–45),[48] and was converted to a standardised score with a mean of 0 and a SD of 1.

*Other adult covariates:* at ages 53, 60–64 and 69, participants reported if they smoked and how many times they had taken part in any sports or vigorous leisure activities in the last month (grouped into >5 times a month, 1–4 times a month or not at all). Adult SEP, assessed by own occupation at age 53 (or at earlier ages if missing at age 53), distinguished the same three groups as for father's occupation.

### Statistical analysis
Stata V.14.2 was used for all analyses. We fitted multilevel models which account for the correlation of repeated measures of grip strength within individuals with age

centred at 53 years. Preliminary multilevel models tested whether adult height was associated with grip strength and remained constant with age,[33] and whether men had stronger grip but a faster rate of decline than women, as expected.[23 28 29 33 38 49 50]

In the main analysis, models were run separately for men and women because of evidence of sex interactions with age and other covariates; these are reported where p<0.1. All models were adjusted for height, with change in grip strength modelled by a linear age term, and with the intercept and slope fitted as random effects. We also assessed whether associations changed with age (ie, whether the risk factor was associated with slope) by testing for interactions between each risk factor and age. We tested for non-linearity with the continuous risk factors (by including a quadratic term). For parsimony, only those age or quadratic terms where p<0.1 remained in the models and are presented in the tables.

To investigate developmental risk factors and grip strength, we first investigated separately the associations with birth weight, physical growth, motor development, cognitive development and childhood SEP. For physical growth, we included all SITAR parameters in the same multilevel models. For motor development, we ran models for age at first sitting, standing and walking, first separately and then mutually adjusted.

Then we took the developmental factors associated with grip strength (p<0.05) and adjusted in turn for each group of adult factors, having shown in supplementary analyses how these adult factors were associated with grip strength. Further details of the models are provided in online supplementary material.

In sensitivity analyses, to assess potential bias introduced by: (a) excluding those participants with no valid observations who were unable to perform the test for health reasons, and (b) mortality or other attrition during follow-up, we reran the multilevel models (1) giving a value representing the midpoint of the lowest sex-specific fifth of grip strength to participants unable for health reasons (n=29, 81 observations) and (2) including binary indicators for mortality (n=287) and attrition (n=601).[27 51]

## Participant and patient involvement

Over the 70 years of the study, the research team has increasingly involved participants, in line with changing norms about conducting cohort studies. Findings from this analysis will be reported with the annual dissemination of study findings in birthday cards that participants have received since age 16, and lay summaries will be placed on the website. Participant involvement includes receiving personal letters from the research team as required, and invitations to participate in birthday celebrations, public engagement activities and focus groups to discuss future data collections. When piloting new questionnaires and assessments, we recruit patients from general practices or from the University College London Hopsitals (UCLH) Patient Public Involvement group, and take into account their feedback when designing the mainstage fieldwork.

# RESULTS
## Characteristics of the sample and preliminary analyses

Mean levels of grip strength, birth weight and adult height were greater in men than women; women had more health conditions and lower SEP than men (table 1). Mean grip strength declined between ages 53 and 69 by 7.5 kg for men and 3.6 kg for women; thus the difference between men and women attenuated over time, although the mean sex difference remained substantial at age 69 (16.1 kg) (table 1). Preliminary multilevel models confirmed that adult height was strongly associated with grip strength (3.2 kg, 95% CI 2.8 to 3.5 per 1 SD increase in height, p≤0.001) and remained constant with age, and that men had stronger grip and a faster rate of decline than women (p<0.001 for sex interaction with age).

## Developmental factors: multilevel models
### Birth weight and physical growth

In models adjusted for adult height and age, there were positive associations between birth weight and grip strength which remained constant at all ages for men (p=0.7 for the birth weight by age interaction) but became weaker with age for women (p=0.05 for the interaction) (table 2A). The association was stronger in men. In men, having a greater weight between birth and 26 years, and a later puberty (as indicated by a positive coefficient for height tempo) was associated with stronger grip. In women, shorter height, greater weight and a slower weight velocity between birth and 26 years were associated with stronger grip (table 2B). These associations did not change with age.

### Motor development

Later ages of attaining infant motor milestones of first sitting and standing were associated with weaker grip; there was an inverse U-shaped relationship between age at first walking and grip strength in men (table 2C-E). These associations remained constant with age as there was no evidence of age interactions. When all three motor milestones, together with height were included together, only the inverse U-shaped relationship between age at first walking and grip strength (for men) and the inverse association between age at first standing and grip strength (for women) remained (online supplementary table 1); so these variables were carried forward.

### Cognitive development

In men, there was an inverse U-shaped relationship between childhood cognitive ability and grip strength at age 53, as indicated by the negative estimates for cognition and cognition squared (table 2F). The positive age interactions with both linear and quadratic cognition terms indicate that the association of childhood cognition with grip strength changes with age. Plotting the estimated mean grip strength for this model for the mean cognitive score, and ±1 SD showed that men of higher childhood cognition had lower grip strength than those with a mean cognitive score at age 53 but a slower decline in grip strength

**Table 1** Characteristics of the sample of 1528 men and 1530 women in the Medical Research Council National Survey of Health and Development with at least one measure of grip strength at ages 53, 60–64 or 69 and known height and birth weight

| | Men | | Women | |
|---|---|---|---|---|
| | N | Mean (SD) or % | N | Mean (SD) or % |
| Grip strength (kg) | | | | |
| 53 years | 1398 | 47.7 (12.2) | 1434 | 27.7 (7.9) |
| 60–64 years | 1003 | 44.6 (11.6) | 1059 | 26.0 (7.4) |
| 69 years | 1036 | 40.2 (8.5) | 1062 | 24.1 (5.8) |
| Physical growth | | | | |
| Birth weight (kg) | 1528 | 3.5 (0.5) | 1530 | 3.3 (0.5) |
| Height growth parameters 2–26 years | 1509 | | 1505 | |
| Height (cm) | | 0.095 (6.1) | | 0.014 (5.8) |
| Height tempo (%) | | −0.079 (6.1) | | −0.19 (7.2) |
| Height velocity (%) | | 0.23 (10.0) | | 0.12 (10.6) |
| Weight growth parameters 0–26 years | 1509 | | 1505 | |
| Weight (kg) | | −0.25 (4.3) | | −0.47 (4.3) |
| Weight tempo (%) | | −0.49 (9.8) | | −0.20 (7.7) |
| Weight velocity (%) | | −1.6 (26.2) | | −0.41 (28.2) |
| Motor development (months) | | | | |
| Age at first sitting | 1414 | 6.6 (1.5) | 1424 | 6.6 (1.5) |
| Age at first standing | 1416 | 11.4 (2.3) | 1419 | 11.3 (2.1) |
| Age at first walking | 1424 | 13.6 (2.5) | 1416 | 13.6 (2.4) |
| Early socioeconomic conditions | | | | |
| Father's occupational class | | | | |
| I and II | 341 | 23.5 | 339 | 23.5 |
| III | 707 | 48.7 | 719 | 49.8 |
| IV and V | 405 | 27.9 | 385 | 26.7 |
| Adult factors | | | | |
| Height (cm) | | | | |
| 53 years | 1428 | 174.7 (6.6) | 1477 | 161.6 (5.9) |
| 60–64 years | 1060 | 174.8 (6.6) | 1148 | 161.6 (5.9) |
| 69 years | 1038 | 173.9 (6.4) | 1079 | 160.6 (6.0) |
| BMI (kg/m$^2$) | | | | |
| 53 years | 1427 | 27.4 (4.0) | 1466 | 27.4 (5.4) |
| 63 years | 1059 | 27.9 (4.1) | 1147 | 27.9 (5.5) |
| 69 years | 1038 | 28.2 (4.6) | 1075 | 28.2 (5.8) |
| Verbal memory (no. words) | | | | |
| 53 years | 1386 | 23.0 (6.2) | 1450 | 24.9 (6.2) |
| 63 years | 1020 | 23.0 (5.9) | 1114 | 25.4 (6.0) |
| 69 years | 1010 | 21.1 (6.0) | 1057 | 23.1 (6.0) |
| No. health conditions 53 years | | | | |
| 0 | 802 | 56.4 | 702 | 47.6 |
| 1 | 448 | 31.7 | 543 | 37.0 |
| 2 | 147 | 10.5 | 159 | 11.3 |
| 3+ | 18 | 1.4 | 57 | 4.1 |
| Smoking status 53 years | | | | |
| Non-smoker or ex-smoker | 1094 | 76.5 | 1150 | 77.7 |

**Table 1** Continued

| | Men | | Women | |
|---|---|---|---|---|
| | N | Mean (SD) or % | N | Mean (SD) or % |
| Smoker | 336 | 23.5 | 330 | 22.3 |
| Smoking status 60–64 years | | | | |
| Non-smoker or ex-smoker | 975 | 87.2 | 1052 | 88.3 |
| Smoker | 136 | 12.2 | 140 | 11.7 |
| Smoking status 69 years | | | | |
| Non-smoker or ex-smoker | 1065 | 89.7 | 1140 | 91.3 |
| Smoker | 122 | 10.2 | 109 | 8.7 |
| Leisure-time physical activity 53 years | | | | |
| Inactive | 675 | 47.2 | 745 | 50.3 |
| Intermediate | 268 | 18.7 | 240 | 16.2 |
| Active | 486 | 34.0 | 495 | 33.4 |
| Leisure-time physical activity 60–64 years | | | | |
| Inactive | 679 | 65.2 | 745 | 62.9 |
| Intermediate | 136 | 13.0 | 240 | 14.4 |
| Active | 227 | 21.8 | 495 | 22.7 |
| Leisure-time physical activity 69 years | | | | |
| Inactive | 654 | 60.2 | 710 | 60.6 |
| Intermediate | 119 | 11.1 | 162 | 13.5 |
| Active | 314 | 28.9 | 256 | 25.9 |
| Qualifications by 26 years | | | | |
| Degree or higher | 211 | 14.6 | 80 | 5.5 |
| 'A-level' or equivalents | 410 | 28.4 | 338 | 23.4 |
| 'O-level' or equivalents | 209 | 14.5 | 371 | 25.7 |
| Less than 'O-level' | 90 | 6.2 | 133 | 9.2 |
| None | 525 | 36.3 | 522 | 36.1 |
| Own occupational class 53 years | | | | |
| I and II | 779 | 51.6 | 555 | 36.6 |
| III | 571 | 37.8 | 641 | 42.2 |
| IV and V | 161 | 10.7 | 322 | 21.2 |
| Died during follow-up | | | | |
| No | 1359 | 88.9 | 1412 | 92.3 |
| Yes | 169 | 11.1 | 118 | 7.7 |
| Other attrition during follow-up | | | | |
| No | 1225 | 80.2 | 1232 | 80.5 |
| Yes | 303 | 19.8 | 298 | 19.5 |

Standardised score for childhood cognitive ability not presented for 1408 men and 1410 women as mean=0 and SD=1.
BMI, body mass index.

(by 0.07 kg/year, 95% CI 0.02 to 0.11 per 1 SD, p=0.003) and thus a higher grip strength at age 69 (figure 1). In women, higher childhood cognitive ability was associated with stronger grip and this remained constant with age.

### Childhood SEP

There was no association between childhood SEP and grip strength in men (table 2G). However, in women there was weak evidence that the association grew stronger with age; women from social classes IV and V showed a faster decline in grip strength (figure 2).

### Mutual adjustment of childhood factors

Mutual adjustment of birth weight, physical growth, age at walking (men), age at standing (women) and cognitive development and childhood SEP identified the factors

**Table 2** Estimates from multilevel models showing mean differences in grip strength (kg) and mean differences in grip strength change (kg/year) for each childhood factor in the Medical Research Council National Survey of Health and Development

| | Men | | | | Women | | | |
|---|---|---|---|---|---|---|---|---|
| | N | Reg. coeff. | 95% CI | P value* | N | Reg. coeff. | 95% CI | P value* |
| **Physical growth** | | | | | | | | |
| A. Birth weight (kg) | 1528 | | | | 1530 | | | |
| Birth weight | | 1.96 | 1.08 to 2.84 | <0.001 | | 1.07 | 0.25 to 1.88 | 0.01 |
| Birth weight×age (year) | | N/A | | | | −0.06 | −0.12 to 0.00 | 0.05 |
| B. Growth parameters | 1509 | | | | 1505 | | | |
| Height size (cm) | | −0.01 | −0.22 to 0.20 | 0.9 | | −0.16 | −0.29 to −0.03 | 0.02 |
| Height tempo (%) | | 0.16 | 0.06 to 0.26 | 0.001 | | −0.02 | −0.04 to 0.07 | 0.5 |
| Height velocity (%) | | −0.02 | −0.10 to 0.05 | 0.5 | | 0.00 | −0.04 to 0.04 | 0.9 |
| Weight size (kg) | | 0.71 | 0.24 to 1.18 | 0.003 | | 0.41 | 0.57 to 0.77 | 0.02 |
| Weight tempo (%) | | −0.06 | −0.12 to 0.01 | 0.08 | | −0.01 | −0.06 0.04 | 0.8 |
| Weight velocity (%) | | −0.07 | −0.14 to 0.00 | 0.06 | | −0.06 | −0.11 to 0.00 | 0.04 |
| **Motor development (months)** | | | | | | | | |
| C. Age at sitting | 1414 | −0.25 | −0.57 to 0.06 | 0.1 | 1424 | −0.18 | −0.38 to 0.01 | 0.07 |
| D. Age at standing | 1416 | −0.27 | −0.47 to −0.07 | 0.008 | 1419 | −0.21 | −0.34 to −0.07 | 0.002 |
| E. Age at walking | 1424 | | | | 1416 | | | |
| Walking | | 1.36 | 0.00 to 2.73 | 0.05 | | −0.13 | −0.25 to −0.01 | 0.03 |
| Walking² | | −0.06 | −0.10 to −0.01 | 0.01 | | N/A | | |
| F. Cognitive development (SD) | 1408 | | | | 1410 | | | |
| Cognition | | −0.46 | −1.10 to 0.17 | 0.1 | | 0.45 | 0.14 to 0.76 | 0.005 |
| Cognition² | | −0.81 | −1.27 to −0.34 | 0.001 | | N/A | | |
| Cognition×age (year) | | 0.07 | 0.02 to 0.11 | 0.003 | | N/A | | |
| Cognition²×age (year) | | 0.03 | 0.00 to 0.06 | 0.08 | | N/A | | |
| G. Father's occupational class | 1453 | | | 0.2 | 1443 | | | 0.4 |
| I and II | | Reference | | | | Reference | | |
| III | | −0.51 | −1.64 to 0.62 | | | −0.24 | −1.15 to 0.67 | |
| IV and V | | −1.29 | −2.58 to 0.00 | | | 0.38 | −0.66 to 1.42 | |
| III×age (year) | | N/A | | | | −0.03 | −0.10 to 0.04 | 0.07 |
| IV×age (year) | | N/A | | | | −0.10 | −0.18 to −0.01 | 0.02 |

All models adjusted for age term and standardised adult height.
*P values are given for the overall tests of associations. P values for sex interactions: birth weight p=0.004; childhood height after adjustment for adult height p=0.07; height tempo p=0.006; age at first waking (quadratic term) p=0.02.
N/A, not available.

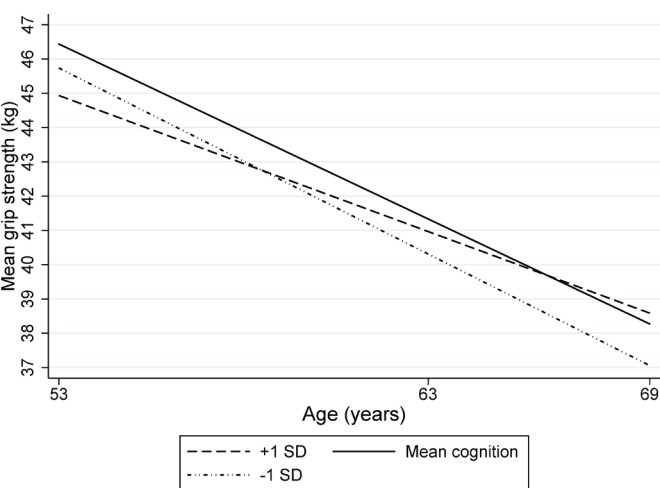

**Figure 1** Mean grip strength (kg) by childhood cognition for men of mean height (based on 1408 men).

most associated with stronger grip. In men, these were higher birth weight and childhood weight, later puberty, the non-linear relationship with age at first walking and the non-linear relationship with childhood cognition which became increasingly linear with age (online supplementary table 2). In women, these were higher birth weight and higher childhood cognition; the estimates for age at first standing were somewhat reduced in this sample with complete childhood data (online supplementary table 3).

### Adult factors: multilevel models

All adult covariates were associated with adult grip strength (online supplementary table 4a-g). Of particular relevance for our hypotheses, higher BMI was associated with stronger grip for men (but not women) and this association levelled off at higher levels of BMI and became weaker with age. Having more health conditions was associated with lower grip strength; for men, this association strengthened with age but for women it weakened. Higher educational levels were associated with stronger grip, especially among women. The association

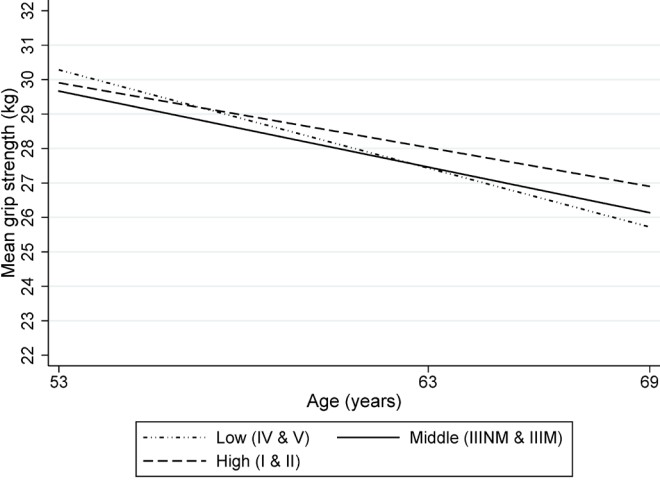

**Figure 2** Mean grip strength (kg) by father's social class for women of mean height (based on 1443 women).

between verbal memory and grip strength was slightly negative in men and not evident in women at age 53 but grew stronger and positive with age.

### Developmental factors adjusted for adult covariates: multilevel models

In men, the estimates for birth weight, height tempo and weight 0–26 years changed little after adjusting in turn for each adult factor with the greatest reduction in the estimate for birth weight occurring when health conditions were included in the model (table 3). In contrast, the increasing association between childhood cognitive ability and grip strength with age was strongly attenuated by including verbal memory and a verbal memory by age interaction in the model (table 4); other adult risk factors had much less impact.

In women, the association between childhood cognitive ability and grip strength was reduced by several adult factors, but especially by educational level (table 5). However, the association between birth weight and grip strength was not reduced by any of the adult factors.

The estimates for age at first walking (in men) and age at first standing (in women) were lower in the sample with complete data; however, these lower estimates were not affected by adjustment for adult factors (online supplementary table 5).

### Sensitivity analyses

Giving a value for grip strength representing the mid-point of the lowest sex specific fifth for those who had no measure because of health reasons (maximum of 81 observations) had either no change or marginal change on the estimates. Compared with participants who were followed up and assessed at age 69, those who died during follow-up had lower mean grip strength at age 53 (−2.4 kg, 95% CI −4.4 to −0.41 for men, p=0.02; and −2.5 kg, 95% CI −3.9 to −1.2 for women, p<0.001) and, for men, there was weak evidence of a faster decline in grip strength (by −0.23 kg/year, 95% CI −0.37 to 0.014, p=0.06). The mean differences in grip strength between those dropping out for reasons other than death and those completing follow-up were smaller (−0.99 kg, 95% CI −2.4 to 0.41 for men, p=0.2; and −1.0 kg, 95% CI −1.9 to −0.16 for women, p=0.02) with no evidence of age interaction. However, estimates for the associations between developmental and adult risk factors and grip strength remained similar after adjusting for death and attrition.

### DISCUSSION

In a prospective, nationally representative British birth cohort, developmental indicators had persisting associations with grip strength over 16 years from midlife to old age. Physical growth, in terms of heavier birth weight and, for men, later puberty and greater weight throughout the growth period was associated with stronger grip, and these effects were robust to adjustment for adult factors. 'On time' motor development for males and advanced motor

**Table 3** Estimates from multilevel models showing mean differences in grip strength (kg) by birth weight, height tempo and weight size in 1295 Medical Research Council National Survey of Health and Development men (2788 observations), adjusted for age term, standardised adult height and all growth parameters and then additionally adjusted for each set of adult factors in turn

| | Birth weight (kg) | | | Height tempo (%) | | | Weight size (kg) | | |
|---|---|---|---|---|---|---|---|---|---|
| | Reg. coeff. | 95% CI | P value | Reg. coeff. | 95% CI | P value | Reg. coeff. | 95% CI | P value |
| Adjusted for age to all growth parameters and adult height | 1.35 | 0.33 to 2.37 | 0.009 | 0.25 | 0.14 to 0.35 | <0.001 | 0.63 | 0.10 to 1.16 | 0.02 |
| Additional adjustments in turn | | | | | | | | | |
| BMI* to BMI², BMI×age | 1.37 | 0.35 to 2.38 | 0.008 | 0.25 | 0.14 to 0.35 | <0.001 | 0.60 | 0.07 to 1.12 | 0.03 |
| Health conditions 53 years Health conditions×age | 1.28 | 0.27 to 2.29 | 0.01 | 0.24 | 0.13 to 0.35 | <0.001 | 0.60 | 0.07 to 1.12 | 0.03 |
| Qualifications | 1.42 | 0.40 to 2.43 | 0.006 | 0.25 | 0.14 to 0.36 | <0.001 | 0.61 | 0.08 to 1.14 | 0.02 |
| Verbal memory* Verbal memory×age | 1.39 | 0.37 to 2.41 | 0.008 | 0.24 | 0.13 to 0.35 | <0.001 | 0.62 | 0.09 to 1.15 | 0.02 |
| Own social class 53 years Own social class×age | 1.42 | 0.40 to 2.43 | 0.006 | 0.24 | 0.14 to 0.35 | <0.001 | 0.60 | 0.07 to 1.12 | 0.03 |
| Smoking and physical activity* | 1.39 | 0.38 to 2.40 | 0.007 | 0.24 | 0.13 to 0.34 | <0.001 | 0.63 | 0.10 to 1.15 | 0.02 |

*Time varying covariates.

development for females were associated with stronger grip which were unexplained by adult factors. Childhood cognitive ability was associated with stronger grip (women) and a slower decline in grip strength (men) during that same life stage; these associations were explained by later education in women and adult cognition in men.

### Comparison with other studies

Comparisons between studies are difficult because different methods have been used to assess decline in grip strength, not all analyses adjust for current adult height and few studies have developmental data. Some studies have only two grip assessments,[25 30 33 34 52] or take the difference between the average of several later assessments from the average of several earlier assessments[32]: both methods are limited in their ability to analyse risk factors related to change. Studies with three assessments used bivariate growth curve models[38] or multilevel models.[27] Other studies with five or more assessments used latent growth curve modelling.[28 53]

**Table 4** Estimates from multilevel models showing mean differences in grip strength (kg) and mean differences in grip strength change (kg/year) by childhood cognition in 1161 Medical Research Council National Survey of Health and Development men (2515 observations), adjusted for age term, standardised adult height and age at first walking and then additionally adjusted for each set of adult factors in turn

| | Childhood cognition (SD) | | | Childhood cognition (SD)²* | | | Childhood cognition (SD)×age (year) | | |
|---|---|---|---|---|---|---|---|---|---|
| | Reg. coeff. | 95% CI | P value | Reg. coeff. | 95% CI | P value | Reg. coeff. | 95% CI | P value |
| Adjusted for age, adult height, walking, walking² | −0.44 | −1.14 to 0.25 | 0.2 | −0.52 | −0.94 to −0.11 | 0.01 | 0.08 | 0.028 to 0.13 | 0.002 |
| Additional adjustments in turn | | | | | | | | | |
| BMI*, BMI² BMI×age | −0.37 | −1.06 to 0.32 | 0.3 | −0.51 | −0.93 to −0.10 | 0.01 | 0.07 | 0.02 to 0.12 | 0.005 |
| Health conditions 53 years Health conditions×age | −0.55 | −1.25 to 0.15 | 0.1 | −0.53 | −0.95 to −0.12 | 0.01 | 0.08 | 0.03 to 0.13 | 0.002 |
| Qualifications | −0.79 | −1.62 to 0.04 | 0.06 | −0.56 | −0.98 to −0.14 | 0.009 | 0.08 | 0.03 to 0.13 | 0.002 |
| Verbal memory* Verbal memory×age | −0.12 | −0.93 to 0.69 | 0.7 | −0.54 | −0.96 to −0.12 | 0.01 | 0.03 | −0.03 to 0.09 | 0.3 |
| Own social class 53 years Own social class×age | −0.80 | −1.59 to 0.00 | 0.05 | −0.52 | −0.94 to −0.10 | 0.01 | 0.07 | 0.02 to 0.13 | 0.01 |
| Smoking and physical activity* | −0.64 | −1.34 to 0.06 | 0.07 | −0.56 | −0.97 to −0.14 | 0.08 | 0.08 | 0.03 to 0.13 | 0.001 |

*Time varying covariates.
BMI, body mass index.

**Table 5** Estimates from multilevel models showing mean differences in grip strength (kg) by birth weight and childhood cognition in 1211 Medical Research Council National Survey of Health and Development women (2709 observations), adjusted for age term and standardised adult height and then additionally adjusted for each set of adult factors in turn

| | Birth weight (kg) | | | Birth weight×age (year) | | | Childhood cognition (SD) | | |
|---|---|---|---|---|---|---|---|---|---|
| | Reg. coeff. | 95% CI | P value | Reg. coeff. | 95% CI | P value | Reg. coeff. | 95% CI | P value |
| Adjusted for age, adult height, age at first standing and mutually adjusted | 1.25 | 0.34 to 2.16 | 0.007 | −0.08 | −0.15 to −0.02 | 0.01 | 0.37 | 0.031 to 0.71 | 0.03 |
| Additional adjustments in turn | | | | | | | | | |
| BMI* | 1.26 | 0.35 to 2.17 | 0.007 | −0.08 | −0.15 to −0.02 | 0.01 | 0.36 | 0.02 to 0.70 | 0.04 |
| Health conditions 53 years* Health conditions×age | 1.27 | 0.37 to 2.19 | 0.006 | −0.09 | −0.15 to −0.02 | 0.01 | 0.27 | −0.07 to 0.61 | 0.1 |
| Qualifications* | 1.22 | 0.31 to 2.13 | 0.008 | −0.08 | −0.15 to −0.02 | 0.01 | 0.019 | −0.4 to 0.48 | 0.9 |
| Verbal memory* Verbal memory×age | 1.29 | 0.38 to 2.20 | 0.005 | −0.09 | −0.16 to −0.02 | 0.008 | 0.22 | −0.17 to 0.61 | 0.3 |
| Own social class | 1.23 | 0.32 to 2.13 | 0.008 | −0.09 | −0.15 to −0.02 | 0.01 | 0.27 | −0.10 to 0.65 | 0.1 |
| Smoking and physical activity* | 1.37 | 0.47 to 2.26 | 0.003 | −0.09 | −0.16 to −0.02 | 0.009 | 0.30 | −0.04 to 0.65 | 0.08 |

*Time varying covariates.
BMI, body mass index.

In NSHD, there was no association between childhood SEP and adult grip strength or its decline after adjusting for developmental factors. Findings from other studies have not been consistent and have adjusted for few, if any, other childhood factors, or have not studied change.[34 36 39 53] The results of a meta-analysis showed modest associations between childhood SEP and adult grip strength at a single time point which were attenuated by adult SEP and current body size, but there was considerable heterogeneity between studies.[54]

The constant effect of birth weight on adult grip strength is consistent with a meta-analysis[55]; this showed a larger estimate for men than women (as this study found) but the sex interaction was not significant. The persisting associations between growth parameters, motor milestones and grip strength are novel findings, and build on previous NSHD work relating to grip strength at age 53,[36] and bone phenotype at age 60–64.[45 56]

The most striking observation in this study was the strengthening of the positive associations between cognition and grip strength with age, whether cognition was assessed in childhood or adult life. This extends an earlier NSHD study showing that the group with meaningful decline in grip strength between ages 53 and 60–64 had lower childhood cognitive ability than those who experienced no meaningful change.[34] In older cohorts, there is growing evidence that changes in grip strength are related to baseline cognition, and that cognitive decline may precede declines in strength,[24] although the few studies investigating covariation in cognition and grip strength have been inconsistent.[38 57] Our findings complement the findings from older cohorts as they cover midlife changes in grip strength over a longer follow-up period than most previous studies.

### Explanation of findings
#### Birth weight, physical growth and motor development
The persistence of the associations between birth weight, growth parameters, motor milestones and grip strength are worth noting given that more proximal factors may come into play as people age which could have diminished this association; however, the size of the estimates suggest that the associations may not be clinically meaningful.

Later puberty (in men) was associated with stronger grip, yet earlier puberty was associated with greater areal and volumetric bone mineral density in this cohort,[45] perhaps due to the differential impact of hormonal regulation. Nevertheless, we found that, controlling for contemporaneous body size, greater weight and slower weight velocity throughout the growth period was associated with both greater grip strength and greater bone size,[44] suggesting an extended growth period may benefit both. This could also be the explanation for the persisting associations between motor milestones and grip strength and, in women, for the inverse association between height during growth and later grip strength, after controlling for adult height.

While the associations between physical growth and grip strength were independent of adult covariates, the number of health conditions attenuated the birth weight effect in men more than other covariates. Lower birth weight is predictive of CVD and diabetes,[58] as is poor muscle strength,[16] and may reflect a common pathway to later disease.

#### Lifetime cognition
Our findings regarding lifetime cognition and grip strength suggest that neural processes have greater

impact on grip strength at older ages than in midlife. The attenuation of the childhood cognitive associations once verbal memory (for men) or education (for women) were taken into account may mean that neurodevelopmental processes play a role in maximising muscle function at maturity but neurodegenerative processes increasingly drive the age-related decline in muscle function. A theoretical model arising out of a review of cognitive ageing, motor learning and motor skills[59] predicts that ageing impairs cognitive functions before affecting the motor systems and that at older ages the connection between cognition and action becomes stronger, as suggested by our findings. To what extent our findings reflect a direct pathway between brain ageing and muscle strength,[60] or shared mechanisms relating, for example, to haemostatic dysregulation or inflammatory processes,[24 61] is yet to be clarified.

### Strengths and limitations

NSHD is one of the very few studies with prospectively assessed factors from development onwards, a wide range of potential covariates and repeat measures of grip strength assessed over a relatively long follow-up period during a critical phase of age-related change. So far, these repeat measures cover midlife to early old age, a period which has been studied less often than later ages. NSHD remains broadly representative of the population born in Britain in the early post war period.[42] One limitation is that it is only possible to model linear change as there are currently only three assessments of grip strength. However, we did investigate whether each association strengthened or weakened with age. Inevitably there were missing data but neither accounting for deaths and attrition, nor including those unable for health reasons, altered our findings. There were also few differences in the size of the estimates in the models using the maximum samples and the models with complete covariate data. We acknowledge that associations that were maintained after multiple adjustments could be due to residual confounding.

### Implications

Muscle strength in later life is dependent on peak muscle function attained by young adulthood, and its subsequent rate of loss. Our findings that developmental as well as adult factors are associated with grip strength from midlife into old age suggest that primary prevention should start early in life and continue across life. Primary prevention should be supported by further research into the determinants of peak muscle function in order to develop the most effective strategies to maximise and maintain function. In a similar vein, a US report suggested it was time to shift the focus on to the primary prevention of osteoporosis, by better understanding the determinants of peak bone mass.[62] Given the associations between lifetime cognition and the level and change in muscle strength, trials to improve childhood or adult cognition should include muscle strength as an additional outcome.

### CONCLUSIONS

Patterns of early growth, attainment of motor milestones and lifetime cognition have persisting associations with grip strength between midlife and old age, even after taking account of adult body size, health conditions and health behaviours. The impact of neural processes strengthened over this stage of life suggesting that at older ages grip strength increasingly reflects both physical and cognitive ageing processes. Interventions across life that promote muscle development or maintain peak muscle strength should increase the chance of an active and independent old age.

**Acknowledgements** The authors would like to thank the Medical Research Council National Survey of Health and Development (NSHD) study members for their lifelong participation and past and present members of the NSHD study team who helped to collect the data.

**Contributors** DK, RC, JMB and RH contributed to the study design and data interpretation and DK, RC and RH collected the data. DK undertook the literature search, the data analysis and wrote the first draft of the manuscript; all authors revised the manuscript. DK is the guarantor and accepts full responsibility for the work and the conduct of the study, had access to the data, and controlled the decision to publish. All authors, external and internal, had full access to all of the data (including statistical reports and tables) in the study and can take responsibility for the integrity of the data and the accuracy of the data analysis.

**Funding** This work was supported by the UK Medical Research Council MC_UU_12019/1, which provides core funding for the MRC National Survey of Health and Development and supports DK, JB, RC and RH by MC_UU_12019/1, MC_UU_12019/2, MC_UU_12019/4. JB also receives support from UCL (Overseas and Graduate Research Scholarships).

**Disclaimer** The funders had no role in the study or the decision to submit the paper for publication.

**Competing interests** All authors have completed the ICMJE uniform disclosure form and declare: DK, JB, RC and RH received financial support from the UK Medical Research Council for the submitted work. JB also receives support from UCL (Overseas and Graduate Research Scholarships); no financial relationships with any organisations that might have an interest in the submitted work in the previous three years; no other relationships or activities that could appear to have influenced the submitted work.

**Patient consent for publication** Not required.

**Ethics approval** Ethical approval for the most recent visit was given by Queen Square Research Ethics Committee (13/LO/1073) and Scotland A Research Ethics Committee (14/SS/1009).

**Provenance and peer review** Not commissioned; externally peer reviewed.

**Data sharing statement** Data used in this publication are available to bona fide researchers on request to the NSHD Data Sharing Committee via a standard application procedure. Further details can be found at http://www.nshd.mrc.ac.uk/data. doi:10.5522/NSHD/Q101; doi:10.5522/NSHD/Q102; 10.5522/NSHD/Q103.

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
