## [Reviewer comments · BMJ Open]

ARTICLE DETAILS

TITLE (PROVISIONAL)	Developmental factors associated with decline in grip strength from midlife to old age: a British birth cohort study
AUTHORS	Kuh, Diana; Hardy, Rebecca; Blodgett, Joanna; Cooper, Rachel

VERSION 1 - REVIEW

REVIEWER	QIANLI XUE Division of Geriatric Medicine and Gerontology, School of Medicine Department of Medicine and the Center on Aging and Health, The Johns Hopkins University, USA
REVIEW RETURNED	29-Aug-2018

GENERAL COMMENTS	General comments: The authors used rich data from a longitudinal birth cohort to study the influence of birthweight and early childhood developmental factors on grip strength trajectory in mid-to-late life, and whether or not such influence was mediated by adult factors. The analysis was thorough and the findings are complex but interesting. However, the manuscript may greatly benefit from better explaining the various regression coefficients from the multilevel model by providing detailed model specifications for the final models in a supplement. It would also be helpful for readers who may not be familiar with the multilevel model to know the exact interpretation of the interaction terms in the tables. For example, in reference to the interaction terms between cognition and age in Table 2f, the statement at the bottom of page 10 “The age interaction terms show that the linear term strengthened...and the quadratic term became weaker with age” is confusing. How could the effect of childhood cognitive ability be “strengthened” by factors (i.e., older age) occurring later in life? It is also perplexing to see the association between higher childhood cognitive ability (as well as higher adult verbal memory) and lower grip strength at age 53. Regarding the missing data issue, it would be a more preferred approach to conduct sensitivity analysis using the complete data rather than including dummy indicators in the model for missingness. It is worth noting that the multilevel model via maximum likelihood estimation is robust to data missing at random. Specific comments:
---

Page 2, lines 33-38: unclear what evidence in supplementary table 4 supports the conclusion that adult verbal memory became increasingly positively associated with grip strength at older ages"? Should the childhood cognitive ability be adjusted in the model in order to test this hypothesis?

Page 4, lines 10-12: need to rephrase the sentence "musculoskeletal disorders rank second in years lived with disability and fourth in disability-adjusted life years" for clarity.

Page 4, line 46: define abbreviation SEP before using it.

Page 7, lines 5-8: what are the measurement units for the measures of childhood cognitive ability? Percentile ranking?

Page 7, lines 9-12: define what you mean by "similar tests," and provide their names and references support the claim that "participants maintained similar ranking across time."

Page 7, line 44: describe how the standardized score for verbal memory was calculated.

Page 10, lines 12-14: Was the p-value <0.001 for sex interaction with age slope? Not with height? Also, need to include in the text the sex-specific estimates of the fixed intercept and age slope. Was the age variable centered around 53 for meaningful interpretation of the intercept?

Page 10, line 22: where is the result from Table 2 supporting the claim that "grip strength ... remained constant with age"? or do you mean the birthweight did not influence the rate of decline in grip strength?

Page 11, lines 3-7: why was "the linear term strengthened" when the main effects of cognition and cognition squared were positive and the interaction terms with age were both negative? Without knowing the magnitude and direction of the fixed effect of the age slope, is it difficult to draw a conclusion such as: "men of higher childhood cognition showed a slower decline in grip strength."

Page 15, lines 48-51: the statement regarding "The attenuation of the childhood cognitive associations once verbal memory or education were taken into account" seems to contradict the increased regression coefficient of -0.79 for men in table 4 for childhood cognition after adjusting for education.

Table 4: how to interpret the increased effect of childhood cognition after adjusting for education in men?

Supplemental table 2: what were included in the "fully-adjusted model"?

REVIEWER	Annie Robitaille Université du Québec à Montréal, Canada
REVIEW RETURNED	18-Sep-2018

GENERAL COMMENTS	In their paper entitled “Developmental factors associated with decline in grip strength from midlife to old age: a British birth cohort study” the authors examine the relationship between grip strength and numerous childhood and adult factors. The large sample size and the longitudinal nature of the data is a clear advantage of the paper. Most importantly, few longitudinal studies have information from birth to older adulthood. All sections of the paper are also well written with no major flaws. My main concern is related to the discussion of the paper and the implications of the research. The authors make a general statement about the implications without going into further detail “Interventions across life that promote muscle development or maintain peak muscle strength should increase the chance of an active and independent old age”. The discussion paper should elaborate on the implications of some of their specific results and differences between men and women. The authors do a good job of comparing their findings with those of previous studies but fail to expand on what their research findings add to our knowledge of physical and cognitive aging across the lifespan and how these can be used to inform policy, programs and practice. Based on their findings, the authors should also elaborate on what researchers needs to focus on in future. The results section could also be better structured with additional sub headings. Given the multiple models and numerous variables that were included, the results are sometimes hard to follow. The adding of subheadings would make it easier for readers. The title of the headings should also be improved. One is called “Developmental risk factors: multilevel models” and then a question is used for the next heading “Which adult factors account for the associations between developmental factors and grip strength?”. Minor changes Define SEP the first time it is mentioned. There is no where that you specify that it’s socio-economic position. Page 8, line 22 & 31, when you write “these are reported where $p < 0.1$”, do you mean $p < 0.01$?
---

VERSION 1 – AUTHOR RESPONSE

Reviewer: 1 (Reviewer Name: QIANLI XUE)

General comments:

The authors used rich data from a longitudinal birth cohort to study the influence of birthweight and early childhood developmental factors on grip strength trajectory in mid-to-late life, and whether or not such influence was mediated by adult factors. The analysis was thorough and the findings are complex but interesting.

RESPONSE: Thank you for these positive comments

However, the manuscript may greatly benefit from better explaining the various regression coefficients from the multilevel model by providing detailed model specifications for the final models in a supplement.

RESPONSE: We have provided detailed model specifications for the final models as part of the Supplemental Material (referred to on page 9, line 3). We have also clarified the meaning of the interactions with age in the text (page 8, line 29).

It would also be helpful for readers who may not be familiar with the multilevel model to know the exact interpretation of the interaction terms in the tables. For example, in reference to the interaction terms between cognition and age in Table 2f, the statement at the bottom of page 10 “The age interaction terms show that the linear term strengthened...and the quadratic term became weaker with age” is confusing. How could the effect of childhood cognitive ability be “strengthened” by factors (i.e., older age) occurring later in life? It is also perplexing to see the association between higher childhood cognitive ability (as well as higher adult verbal memory) and lower grip strength at age 53.

RESPONSE: A significant age interaction term for both childhood cognition and childhood cognition squared means that these associations change with age. In these circumstances, we have explained the interpretation in further detail, aided by plotting out the estimates for mean grip strength from the model (Figure 1). Thus, the estimates from the multilevel model for childhood cognition, childhood cognition² and their respective age interaction terms with grip strength in men in Table 2f were used to plot the mean grip strength for men with mean levels of the cognitive score and with scores either 1 SD above or below. These model estimates show that men with a 1 SD higher cognitive score had a slower rate of decline in grip strength than those with a mean score and those with 1SD lower scores had the fastest grip strength decline. We have modified the sentence to clarify this (page 11, lines 17-33).

Regarding the missing data issue, it would be a more preferred approach to conduct sensitivity analysis using the complete data rather than including dummy indicators in the model for missingness. It is worth noting that the multilevel model via maximum likelihood estimation is robust to data missing at random.

RESPONSE: As the reviewer rightly says, multilevel models are robust to data missing at random (MAR). We feel that our sensitivity analysis is more appropriate than a complete data analysis, which would estimate effects only in a healthy survivor group. The approach we used has been discussed and applied in the ageing field, for example in the papers we already referenced to justify this approach. Note that we assessed the interaction between the missingness variable and age which showed that those dropping out had lower grip strength on average than those remaining in the study, although there was no or weak evidence of a difference in change with age.

Specific comments:

Page 2, lines 33-38: unclear what evidence in supplementary table 4 supports the conclusion that adult verbal memory became increasingly positively associated with grip strength at older ages”? Should the childhood cognitive ability be adjusted in the model in order to test this hypothesis?

RESPONSE: We have simplified this sentence in the abstract by removing the last phrase ‘which became increasingly positively associated with grip strength at older ages’ as adult verbal memory is not the focus of the paper but is included to test whether it explains the associations between developmental factors and grip strength. The inclusion of Supplementary Table 4 is to enable the reader to see how the adult risk factors are associated with grip strength before we use them as covariates for Tables 3-5. Estimates from the model presented in Supplementary Table 4 show that higher verbal memory was associated with lower grip strength at age 53 [regression coefficient (95%

CI) kg per SD cognition: -0.67 (-1.24, -0.093)]. The positive interaction between verbal memory and age [0.10 (0.061, 0.15)] indicates that the decline in grip strength is less in those with higher cognition – or equivalently that the association becomes less negative with increasing age. Estimating the association at age 69 indicates that the association becomes positive at older ages (the estimate at 69 is: $-0.67 + [0.10 * 16\text{yrs}] = 0.93$ kg per SD cognition), such that higher verbal memory is now associated with higher grip strength.

Page 4, lines 10-12: need to rephrase the sentence “musculoskeletal disorders rank second in years lived with disability and fourth in disability-adjusted life years” for clarity.

RESPONSE: We have clarified and simplified this sentence as follows: “Worldwide, musculoskeletal disorders rank second in terms of their impact on years lived with disability.” (page 4, line 12)

Page 4, line 46: define abbreviation SEP before using it.

RESPONSE: This has now been done (page 4, line 49).

Page 7, lines 5-8: what are the measurement units for the measures of childhood cognitive ability? Percentile ranking?

RESPONSE: Childhood cognitive ability scores were standardised and so a unit is one standard deviation. This has been further clarified on page 7, first paragraph.

Page 7, lines 9-12: define what you mean by “similar tests,” and provide their names and references support the claim that “participants maintained similar ranking across time.”

RESPONSE: Where missing at age 15 years, cognitive scores were imputed using standardised scores from similar cognitive tests undertaken at ages 11 years or 8 years (if missing at 15 and 11). All of these tests of verbal and nonverbal ability at each age were developed by the National Foundation for Educational Research. At age 8, there were tests of reading comprehension (sentence completion), pronunciation, vocabulary, and nonverbal reasoning. At age 11, there were tests of verbal and nonverbal intelligence (series completion), arithmetic (addition, multiplication, subtraction, and division), pronunciation (as at age 8), and vocabulary (as at age 8). We justified imputing standardised scores at 8 or 11 if missing at 15 years based on the observation that participants in the analytical sample with cognitive test scores at more than one age in childhood maintained a similar ranking over time. For example, we used this method in Cooper et al 2017 [ref 34] which we now cite again at the end of the sentence on page 7. In that paper, we state that the Pearson correlation coefficients between the cognitive score at age of 15 years and those at ages of 11 and 8 years were 0.87 and 0.72, respectively and so this is now summarised on page 7, line 8.

Page 7, line 44: describe how the standardized score for verbal memory was calculated.

RESPONSE: The scores for the three trials of the 15-item word list task were summed and then standardised to have a mean of zero and an SD of one. We have added a phrase to clarify this (page 7, line 44).

Page 10, lines 12-14: Was the p-value <0.001 for sex interaction with age slope? Not with height? Also, need to include in the text the sex-specific estimates of the fixed intercept and age slope. Was the age variable centered around 53 for meaningful interpretation of the intercept?

RESPONSE: Thanks for spotting this mistake. The p-value does indeed refer to the sex interaction with age and this has now been corrected (page 10, line 28-33). We also clarified that we fitted multilevel models which account for the correlation of repeated measures of grip strength within individuals with age centred at 53 years.

Page 10, line 22: where is the result from Table 2 supporting the claim that “grip strength ... remained constant with age”? or do you mean the birthweight did not influence the rate of decline in grip strength?

RESPONSE: Birth weight did not influence the rate of decline in men – this is equivalent to saying the association is the same at all ages. However, on re-checking our figures we found that the birthweight and grip association in women weakened with age; we have amended the relevant text and Tables 2a, Table 5 and Supplementary Tables 3 and 5 and added (page 10, line 26-33) the following... “In models adjusted for adult height and age, there were positive associations between birthweight and grip strength which remained constant at all ages for men (p-value =0.7 for the birthweight by age interaction) but became weaker with age for women (p-value=0.05 for the interaction) (Table 2a). The association was stronger in men.”

Page 11, lines 3-7: why was “the linear term strengthened” when the main effects of cognition and cognition squared were positive and the interaction terms with age were both negative? Without knowing the magnitude and direction of the fixed effect of the age slope, is it difficult to draw a conclusion such as: “men of higher childhood cognition showed a slower decline in grip strength.”

RESPONSE: As we outlined in response to the same comment made above under General comments, we agree that it is difficult to interpret the individual coefficients here and that is why we present a figure of the estimated mean effects on grip strength. The age interaction terms indicate that childhood cognition is associated with the slope and plotting the estimated means for different levels of cognition shows that men of higher childhood cognition exhibit a slower decline in grip strength (Figure 1).

Page 15, lines 48-51: the statement regarding “The attenuation of the childhood cognitive associations once verbal memory or education were taken into account” seems to contradict the increased regression coefficient of -0.79 for men in table 4 for childhood cognition after adjusting for education.

RESPONSE: Thank you for spotting a potentially confusing sentence. We have now modified the sentence to show the difference for men and women i.e. “The attenuation of the childhood cognitive associations once verbal memory (for men) or education (for women) were taken into account suggests...”

Table 4: how to interpret the increased effect of childhood cognition after adjusting for education in men?

RESPONSE: It should be noted that there is only an increase in the negative linear component of the association between childhood cognition and grip strength after adjustment for education, but little impact on the quadratic (see figure below). Adjustment for education changes the shape of the inverse U-shaped association because those with higher cognition are more likely to have higher levels of education while higher education is associated with higher grip strength; thus, the association at the top end of the distribution becomes more negative. We feel that this change is not large enough to warrant comment as the main conclusion is that education does not explain the association.

Figure: Mean grip strength by childhood cognition, before and after adjustment for education

Supplemental table 2: what were included in the “fully-adjusted model”?

RESPONSE: We have now made it clearer that all the variables listed were included in the fully adjusted model plus the age term and height.

Reviewer: 2 (Reviewer Name: Annie Robitaille)

In their paper entitled “Developmental factors associated with decline in grip strength from midlife to old age: a British birth cohort study” the authors examine the relationship between grip strength and numerous childhood and adult factors. The large sample size and the longitudinal nature of the data is a clear advantage of the paper. Most importantly, few longitudinal studies have information from birth to older adulthood. All sections of the paper are also well written with no major flaws.

RESPONSE: Thank you for these positive comments.

My main concern is related to the discussion of the paper and the implications of the research. The authors make a general statement about the implications without going into further detail “Interventions across life that promote muscle development or maintain peak muscle strength should increase the chance of an active and independent old age”. The discussion paper should elaborate on the implications of some of their specific results and differences between men and women.

The authors do a good job of comparing their findings with those of previous studies but fail to expand on what their research findings add to our knowledge of physical and cognitive aging across

the lifespan and how these can be used to inform policy, programs and practice. Based on their findings, the authors should also elaborate on what researchers needs to focus on in future.

RESPONSE: We have now slightly reordered the discussion with subheadings and include a subheading called ‘Implications’. Here (page 17, line 31-57) we discuss the implications of our study for future research and policy.

The results section could also be better structured with additional sub headings. Given the multiple models and numerous variables that were included, the results are sometimes hard to follow. The adding of subheadings would make it easier for readers. The title of the headings should also be improved. One is called “Developmental risk factors: multilevel models” and then a question is used

for the next heading “Which adult factors account for the associations between developmental factors and grip strength?”

RESPONSE: We have added additional subheadings in the results and the discussion which we hope makes both sections easier to follow. (pages 10, lines 24 & 47; page 11, lines 12, 38 & 47; page 12, lines 10 & 31; page 14, line 12; page 15, line 33 & 36; page 16, line 22 & 54; page 17, line 31)

Minor changes

Define SEP the first time it is mentioned. There is no where that you specify that it's socio-economic position.

RESPONSE: Apologies for this omission. We now spell out SEP the first time it is mentioned (page 4, line 49).

Page 8, line 22 & 31, when you write “these are reported where $p < 0.1$ ”, do you mean $p < 0.01$?

RESPONSE: We report interactions if $p < .1$ because of the lower power of such interactions. The cut-point of $p < 0.1$ is already outlined on page 8, line 24.